# Multidrug-resistant bacterial profiles of inanimate objects at Zewditu Memorial Hospital, Addis Ababa, Ethiopia

Hiwot Lidet Yosef *, Meron Yohannes, Regasa Diriba, Melese Hailu Legese

Department of Medical Laboratory Sciences, College of Health Sciences, Addis Ababa University, Addis Ababa, Ethiopia

* halleljezu@gmail.com

## Abstract

### Introduction

Nosocomial pathogens are responsible for severe infections and are typically acquired in healthcare settings within days of patient admission. One of the primary contributors to the spread of healthcare-associated infections (HAIs) is the transfer of bacteria from inanimate surfaces to patients.

### Methods

A cross-sectional study was done at Zewditu Memorial Hospital from June to November 2024. For this study, 204 inanimate surfaces from operating rooms and intensive care units were swabbed and cultured on MacConkey and blood agar. Bacteria were identified based on their colony morphology, gram staining, and conventional biochemical tests. Antimicrobial susceptibility testing was done using disk diffusion. Extended-spectrum beta-lactamase (ESBL) producing gram-negative bacteria were confirmed phenotypically using the double disc synergy and combination disc methods, while carbapenemase producers were identified using the Modified Carbapenem inactivation method. Methicillin-resistant *Staphylococcus aureus* (MRSA) was detected via the Cefoxitin Disk Diffusion Test. A P-value of less than 0.05 was considered statistically significant. Data were analyzed using SPSS version 20.

### Results

Of the 204 swabbed samples, 77.45% (n = 158/204) showed bacterial growth, yielding a total of 171 bacterial isolates. Among these, Gram-positive bacteria comprised 71.3% (n = 122/171), while Gram-negative bacteria accounted for 28.7% (n = 49/171). The most prevalent isolates were Coagulase-negative *Staphylococcus* (CoNS), making up 46.1% (n = 79/171), followed by *Bacillus* spp. at 21.6% (n = 37/171). Out of the total isolates, 55 were identified as pathogenic bacteria based on their

**Data availability statement:** All relevant data are within the paper and its Supporting Information files.

**Funding:** The author(s) received no specific funding for this work.

**Competing interests:** No authors have competing interests.

**Abbreviations:** CDC, Centers for Disease Control and Prevention; CLSI, Clinical Laboratory Standard Institute; CONS, Coagulase-Negative Staphylococcus; ESBL, Extended Spectrum β-Lactamase; GNB, Gram-Negative bacteria; GPB, Gram-Positive bacteria; HCWs, Healthcare workers; HCAI, Health care-associated infection; HAI, Hospital Acquired Infection; ISO, International Organization for Standardization; MRSA, Methicillin Resistant Staphylococcus aureus; MDR, Multi-Drug Resistance; MHA, Mueller Hinton agar; ICUs, Intensive Care Units; ORs, Operation Rooms; SPSS, Statistical Package for Social Science.

potential to cause disease and selected for antibiotic resistance testing. *P. aeruginosa* 36.7% (n = 18/49), *Acinetobacter* spp. 16.3% (n = 8/49), *E. coli* 14.2% (n = 7/49) and *S. aureus* 4.9% (n = 6/122) were the commonest pathogenic bacteria identified. Gram-negative bacteria showed high resistance to ampicillin (67.3%), amoxicillin and clavulanic acid (61.2%), ciprofloxacin (63.2%), sulfamethoxazole-trimethoprim (63.2%), cefepime (57.1%), and piperacillin-tazobactam (55.1%). Similarly gram-positive bacteria showed high resistance to azithromycin (100%), penicillin (100%), clindamycin (100%), and erythromycin (100%). Multidrug resistance was observed in 61.8% (n = 34/55) of the tested gram-negative and gram-positive isolates. The incidence of ESBL-producing and carbapenemase-producing bacteria among the suspected gram-negative bacteria was 26.5% (n=13/49) and 12.2% (n=6/49), respectively. The prevalence of MRSA was 50% (n=3/6).

## Conclusion

The study identified a significant presence of multidrug-resistant bacteria in the hospital environment and on inanimate surfaces, emphasizing the urgent need for effective infection prevention and control measures.

## Introduction

Nosocomial pathogens are microorganisms that cause hospital-acquired infections (HAIs), which are contracted in healthcare settings [1]. Healthcare-associated infections pose significant risks to both patient and public health, leading to higher morbidity rates, prolonged hospital stays, and increased mortality [2]. Factors such as inadequate infection prevention and hygiene practices, improper equipment sterilization, and the emergence of resistant bacterial strains contribute to the spread of HAIs [3]. Contaminants from inanimate objects and surfaces in intensive care units (ICUs) and operating rooms (ORs) exacerbate patient mortality and morbidity, as patients in these areas are particularly vulnerable to infections [2].

The ability of bacteria to spread from inanimate objects to humans depends on several factors, including the type of microorganism, the size of the inoculum, the environment's temperature and humidity, the porosity of the object, the presence of organic matter, the microorganism's ability to form biofilms, and the effectiveness of infection control measures [4]. Both gram-positive and gram-negative bacteria have been found on inanimate surfaces. These bacteria can survive for several months on dry surfaces, but their survival rate increases in humid and cooler environments [2]. Bacteria such as *Klebsiella pneumoniae*, *Escherichia coli*, *Staphylococcus aureus*, *Proteus* spp., *Pseudomonas aeruginosa*, and *Enterococcus* spp. are major contributors to healthcare-acquired infections and are widely recognized as potentially deadly pathogens in hospital settings [5,6].

Gram-negative bacteria (GNB) pose a significant global health threat due to their ability to produce beta-lactamase and carbapenemase, which contribute to antimicrobial resistance [7]. Bacteria producing extended-spectrum beta-lactamase

(ESBL) enzymes are responsible for over 19% of nosocomial infections [8]. These enzymes can hydrolyze penicillin, monobactams, and third-generation cephalosporins, while carbapenemase and metallo-beta-lactamase enzymes can break down carbapenem medications [9]. Common ESBL- and carbapenemase-producing bacteria in nosocomial infections include *Escherichia coli*, *Klebsiella pneumoniae*, *P. aeruginosa*, and *Enterobacter cloacae* [10]. The emergence of novel beta-lactamases in gram-negative bacteria, capable of breaking down cephalosporins and carbapenems, is an alarming development. Infections caused by these organisms increase treatment costs, hospital stays, morbidity, and mortality rates [11].

Penicillin-resistant pneumococci and methicillin- and vancomycin-resistant *Staphylococcus aureus* are the most common drug-resistant gram-positive bacteria found in healthcare settings [5]. Another bacterium associated with healthcare-associated infections and environmental contamination is *Acinetobacter baumannii* [12].

The most frequent hospital-acquired infections from ICUs and ORs include bloodstream, urinary tract, surgical site, and respiratory infections [3]. The risk of these infections is heightened by inadequate cleanliness, lack of routine infection prevention measures, insufficient control procedures, and unsafe medical environments [13]. Particularly in developing countries, there is a lack of information on the extent and types of contamination, as well as the microbiological profiles of commonly used medical instruments and inanimate surfaces in hospitals. Therefore, ongoing research in this area is necessary to address and mitigate the problem.

## Methods and materials

From June to November 2024, a hospital-based cross-sectional study was conducted at Zewditu Memorial Hospital, located in the Kirkos sub-city, Addis Ababa, Ethiopia. The hospital has a capacity of 231 beds, 11 wards, and 3 emergency wards and it provides medical care to approximately 115,102 patients annually. Swab specimens were collected from adult and neonatal intensive care units (ICUs), the gynecology and pediatric neurology (CDC operation rooms), and the cesarean section units (delivery units). A total of 204 swabs were collected from selected equipment and environments of those operating rooms and intensive care units. All swab samples were collected during the morning after cleaning had been performed and before staff began their daily activities. Sterile swabs, moistened with 85% normal saline, were used to sample high-touch surfaces after careful observation. Swabbing was performed in parallel spaced stripes with slight rotation, followed by perpendicular stripes, in accordance with ISO/DIS 14698−1 guidelines [14]. Each sample was then transferred into a leak-proof container containing tryptone soya broth as a transport medium.

### Bacterial culture, identification, and interpretation of culture results

All swab samples were inoculated onto MacConkey agar and Blood agar and incubated aerobically at 37°C for 18–24 hours. Each culture-positive bacteria were identified based on their colony morphology, Gram staining, and biochemical characteristics. Biochemical tests used to identify Gram-negative bacteria were the indole, triple sugar iron agar, urea utilization, citrate utilization, mannitol, lysine iron agar decarboxylation, and oxidase. On the other hand, Gram-positive bacteria were identified using Gram staining, catalase, and coagulase tests, with results interpreted according to laboratory standard operating procedures.

### Antimicrobial susceptibility testing

Antimicrobial susceptibility testing was conducted using disc diffusion following the Clinical and Laboratory Standards Institute (CLSI) guidelines. After placing selected antibiotics on a Muller Hinton Agar (MHA) plate inoculated with the test bacteria, the plate was incubated for 16–18 hours. All Gram-negative bacteria were tested against amikacin, amoxicillin-clavulanic acid (AMC), gentamicin, ampicillin, cefotaxime, ceftazidime, ceftriaxone, chloramphenicol, ciprofloxacin, cefepime, ertapenem, imipenem/meropenem, trimethoprim-sulfamethoxazole (SXT), and piperacillin-tazobactam.

Gram-positive bacteria were tested against gentamicin, ciprofloxacin, cefoxitin, trimethoprim-sulfamethoxazole (SXT), penicillin, azithromycin, clindamycin, erythromycin, and tetracycline. Zones of inhibition were measured and interpreted according to the CLSI 34th edition (2024) [15].

### Extended-spectrum β-lactamase (ESBL) detection and confirmation

Gram-negative bacteria that were suspected for Extended-spectrum β-lactamase (ESBL) production were confirmed using the double-disk synergy test and combination disc test. Using the double-disk synergy test, a disc containing amoxicillin-clavulanate (20 μg/10 μg) (augmentin) and a 30-μg disc containing each third-generation cephalosporin test antibiotic were compared for synergy. The discs were positioned 20 mm apart from one another on an MHA that had been swabbed with the test isolate. A clear enlargement of the margin of the cephalosporin inhibition area near the augmentin disc was regarded as a positive ESBL test result. While using the combination disk test, ceftazidime (30 μg) disks were used alone and in conjunction with clavulanic acid (30 μg/10 μg) to confirm the existence of ESBLs. A zone diameter ≥5 mm increase with ceftazidime/clavulanate disks than in ceftazidime disks was found to produce ESBL [15].

### Carbapenemase detection and confirmation

Suspected carbapenemase-producing gram-negative bacteria were confirmed phenotypically using the modified carbapenem inactivation method. Briefly, the method consists of suspending a 1 μl loop of test organisms in 2 mL of trypticase soy broth. A 10 μg meropenem disc was added to the prepared suspension and incubated for 4 hours at 35 °C. A suspension of *E. coli* ATCC 25922 calibrated to 0.5 McFarland was prepared and inoculated on a Mueller-Hinton agar plate. After 4 hours, the meropenem disc was removed and inserted on an MHA plate previously inoculated with the indicator strain *E. coli* ATCC 25922. The plates were then incubated for 18–24 h at 35 °C, and the plates were analyzed based on the Clinical and Laboratory Standards Institute's recommendations [15].

### Methicillin-resistant *Staphylococcus aureus* (MRSA) screening and confirmation

Methicillin-resistant *Staphylococcus aureus* (MRSA) was identified using the cefoxitin disk diffusion test. Isolate was initially grown on a Mueller-Hinton agar plate using a 30-μg cefoxitin disk and a 0.5 McFarland standard suspension. Every plate was incubated aerobically for the entire night at 37°C. The test was performed according to CLSI guidelines [15].

### Data quality assurance

The reliability of the research results was ensured through rigorous quality assurance measures throughout data collection and laboratory procedures. Strict adherence was maintained while collecting, labeling, handling, and transporting biological samples as well as preparing media according to manufacturer guidelines and laboratory Standard Operating Procedures (SOPs). The media's quality was verified against CLSI standards, ensuring compliance with expiration dates. To minimize contamination, aseptic techniques were employed during sample collection and inoculation onto culture media. Additionally, all pre-analytical procedures adhered to standard operating guidelines. The sterility and performance of the culture media were assessed by incubating them overnight at 37°C. The performance of MacConkey agar and blood agar plates was evaluated using control strains, including *Escherichia coli* (ATCC 25922), *Proteus mirabilis* (ATCC 35659), *Staphylococcus aureus* (ATCC 25923), and *Streptococcus pneumoniae* (patient strain). Antibiotic efficacy testing was performed using international control bacterial strains such as *E. coli* (ATCC 25922), *S. aureus* (ATCC 25923), and *Pseudomonas aeruginosa* (ATCC 27853). Additionally, biochemical test media were inoculated with known positive and negative bacterial controls to verify accuracy. All isolated bacteria were stored in accordance with the laboratory's standard operating procedures.

### Data analysis and interpretation

Data was analyzed using SPSS version 20.0 to show the incidence of bacterial pathogens on inanimate surfaces as well as their antibiotic resistance trend. The findings were evaluated via descriptive statistics, and the findings were described using tables, graphs, and text.

### Ethics approval

The study was approved by the department of research and ethics review committee of the medical Laboratory Sciences, College of Health Sciences; Addis Ababa University (Ref. No. MLS/165/24) and City Government of Addis Ababa health bureau (Ref. No. A/A/H/10432/227). Letter of permission was obtained from Zewditu Memorial Hospital to access wards and collect data and Tikur Anbessa Specialized Teaching Hospital to use the necessary materials and to conduct laboratory analysis.

### Results

#### Prevalence of bacterial isolates from inanimate surfaces and medical equipment at ICUs and ORs

In this study, a total of 204 swab samples were collected from various inanimate objects in the operating rooms (ORs) (n=113) and intensive care units (ICUs) (n=91) of Zewditu Memorial Hospital. Among all cultured samples, the overall bacterial prevalence was 77.5% (n=158/204). A total of 171 bacterial isolates were identified from the collected swabs. Of the 171 bacterial isolates, 71.3% (n=122/171) were Gram-positive bacteria, while Gram-negative bacteria accounted for 28.7% (n=49/171) (Fig 1).

Gram-positive bacteria identified in this study included *Staphylococcus aureus* 4.9% (n=6/122), Coagulase-negative *Staphylococci* (CoNS) 64.7% (n=79/122), and *Bacillus* spp. 30.3% (n=37/122). Among the Gram-negative bacteria, the

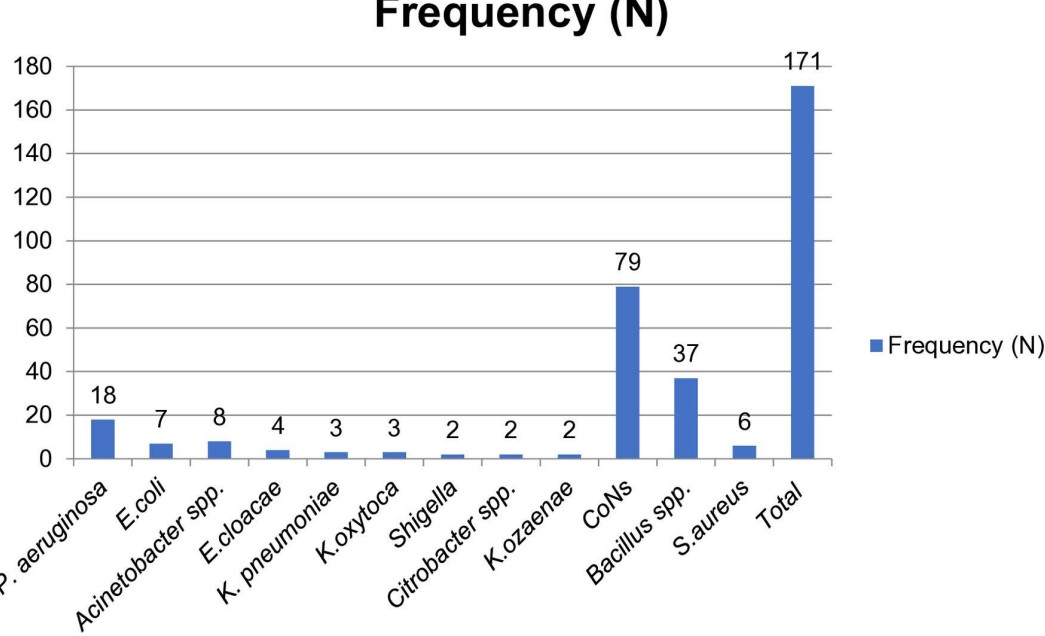

**Fig 1. Gram negative and gram positive bacteria isolated from environmental surfaces and medical equipment at Zewditu Memorial Hospital.**
N-number of bacteria; CoNs-coagulase negative *Staphylococcus aureus.*

most common isolates were *Pseudomonas aeruginosa* 36.7% (n=18/49), *Acinetobacter* spp. 16.3% (n=8/49), and *Escherichia coli* 14.2% (n=7/49). Overall, CoNS was the most frequently isolated bacterium, accounting for 46.1% (n=79/171), followed by *Bacillus* spp. at 21.6% (n=37/171) and *Pseudomonas aeruginosa* at 10.5% (n=18/171).

## Distribution of bacterial isolates between intensive care units and operation rooms

Of all 171 bacteria, 50.8% (n=87/171) were identified in intensive care units (ICUs). Both Gram-positive and Gram-negative bacteria showed variations across different wards, with operation rooms (ORs) reporting 71.4% (n=60/84) Gram-positive and 28.5% (n=24/84) Gram-negative bacteria, while ICUs had 71.2% (n=62/87) Gram-positive and 28.7% (n=25/87) Gram-negative bacteria.

In ICUs, Gram-positive bacteria were predominantly responsible for contamination, comprising 71.2% (n=62/87). Coagulase-negative *Staphylococci* (CoNS) were the most common bacteria identified at ICU, accounting for 52.8% (n=46/87), followed by *Bacillus* spp. at 12.6% (n=11/87). The majority of bacterial isolates in ICUs were from the neonatal intensive care unit (NICU), which accounted for 51.7% (n=45/87) of the cases. Within the NICU, the leading pathogens were CoNS from Gram-positive bacteria and *Pseudomonas aeruginosa* from Gram-negative bacteria, each with isolation rates of 51.1% (n=23/45) and 13.3% (n=6/45), respectively. In the operation rooms (ORs), contamination was primarily caused by Gram-positive bacteria, which made up 71.4% (n=60/84). The most common pathogens in the ORs were CoNS at 39.2% (n=33/84) followed by *Bacillus* spp. at 30.9% (n=26/84) (Table 1).

## Distribution of bacterial pathogens over different surfaces

The highest number of bacterial-contaminated samples were collected from environmental surfaces (14%), followed by tables (12.8%) and patient monitors (9.94%). Environmental surfaces were predominantly contaminated with *Bacillus* spp. 37.5% (n = 9/24), followed by Coagulase-negative *Staphylococci* (CoNS) (29.1%, n = 7/24). Tables used by healthcare workers in operating rooms and intensive care units were primarily contaminated with CoNS 45.4% (n = 10/22). Patient monitors were mainly colonized by CoNS 47% (n = 8/17), *Bacillus* spp. 23.5% (n = 4/17), and *Acinetobacter* spp. 11.7% (n = 2/17) (Table 2).

**Table 1. Distribution of bacteria among wards isolated from environmental surfaces and medical equipment at Zewditu Memorial Hospital.**

| Bacterial isolates | Adult ICUs n(%) | Neonatal ICUs n(%) | Major OR n(%) | CS OR n(%) | Gyn & Pedi OR n(%) | Total n(%) |
|---|---|---|---|---|---|---|
| *CoNS* | 23(29.11) | 23(29.11) | 18(22.7) | 12(15.18) | 3(3.79) | 79(46.1) |
| *Bacillus spp.* | 5(13.5) | 6(16.2) | 15(40.5) | 9(24.3) | 2(5.4) | 37(21.6) |
| *S. aureus* | 3(50) | 2(33.33) | 0(0) | 1(16.66) | 0(0) | 6(3.5) |
| *P. aeruginosa* | 3(16.66) | 5(27.77) | 0(0) | 0(0) | 10(55.55) | 18(10.5) |
| *Klebsiella spp.* | 2(11.11) | 4(22.2) | 1(12.5) | 0(0) | 1(12.5) | 8(4.67) |
| *E. coli* | 2(28.5) | 1(14.28) | 1(14.28) | 1(14.28) | 3(42.8) | 7(4.09) |
| *Acinetobacter* spp. | 2(25) | 0(0) | 2(25) | 4(50) | 0(0) | 8(4.67) |
| *Citrobacter* spp. | 0(0) | 1(50) | 1(50) | 0(0) | 0(0) | 2(1.16) |
| *E. cloacae* | 1(25) | 2(50) | 0(0) | 0(0) | 1(25) | 4(2.33) |
| *Shigella* spp. | 1(50) | 1(50) | 0(0) | 0(0) | 0(0) | 2(1.16) |
| Total | 42(24.5) | 45(26.3) | 38(22.2) | 27(15.7) | 19(11.1) | 171(100) |

n- number of tested strains; %- percentage; ICU-intensive care unit; CoNS- Coagulase-negative *Staphylococci*; Gyn & Pedi OR- (gyneacology & pediatrics neurology operation room); CSOR- cesarean section operation room.

**Table 2. Distribution of bacterial pathogens over different surfaces isolated from environmental surfaces and medical equipment at Zewditu Memorial Hospital.**

| Inanimate objects | CoNS n(%) | Bacillus spp. n(%) | S. aureus n(%) | P. aeruginosa n(%) | Klebsiella spp. n(%) | E. coli n(%) | Acinetobacter spp. n(%) | Citrobacter spp. n(%) | E. cloacae n(%) | Shigella spp. n(%) | Total n(%) |
|---|---|---|---|---|---|---|---|---|---|---|---|
| Anastasia machine (7) | 3(3.79) | 1(2.7) | 0(0) | 1(5.5) | 0(0) | 1(14.28) | 1(1.25) | 0(0) | 0(0) | 0(0) | 7(4.09) |
| Bed (11) | 8(10.1) | 1(2.7) | 0(0) | 0(0) | 2(25) | 0(0) | 0(0) | 0(0) | 0(0) | 0(0) | 11(6.4) |
| Linens (15) | 6(7.5) | 4(10.8) | 2(33.3) | 1(5.5) | 2(25) | 0(0) | 0(0) | 0(0) | 0(0) | 0(0) | 15(8.77) |
| Monitor (17) | 8(10.1) | 4(10.8) | 0(0) | 1(5.5) | 0(0) | 1(14.2) | 2(25) | 1(50) | 0(0) | 0(0) | 17(9.94) |
| Suction Machine (8) | 3(3.7) | 2(5.4) | 0(0) | 0(0) | 0(0) | 1(14.2) | 0(0) | 0(0) | 1(25) | 1(50) | 8(4.67) |
| OR Table (6) | 3(3.7) | 3(8.1) | 0(0) | 0(0) | 0(0) | 0(0) | 0(0) | 0(0) | 0(0) | 0(0) | 6(3.5) |
| Tables (22) | 10(12.6) | 4(10.8) | 0(0) | 5(27.77) | 0(0) | 3(42.8) | 0(0) | 0(0) | 0(0) | 0(0) | 22(12.8) |
| Work Station (6) | 3(3.79) | 0(0) | 0(0) | 2(11.11) | 1(12.5) | 0(0) | 0(0) | 0(0) | 0(0) | 0(0) | 6(3.5) |
| Oxygen Cylinder (12) | 7(8.8) | 3(8.1) | 0(0) | 1(5.5) | 0(0) | 1(14.28) | 0(0) | 0(0) | 0(0) | 0(0) | 12(7.01) |
| Bed Trails (10) | 6(7.59) | 1(2.7) | 1(16.66) | 0(0) | 1(5.55) | 0(0) | 1(12.5) | 0(0) | 0(0) | 0(0) | 10(5.8) |
| OR light (11) | 5(6.3) | 3(8.1) | 0(0) | 2(11.1) | 0(0) | 0(0) | 1(12.5) | 0(0) | 0(0) | 0(0) | 11(6.4) |
| IV Stand (12) | 6(7.5) | 1(2.7) | 1(16.66) | 3(16.66) | 0(0) | 0(0) | 1(12.5) | 0(0) | 0(0) | 0(0) | 12(7.01) |
| Environmental surfaces (24) | 7(8.8) | 9(24.3) | 2(33.33) | 2(11.11) | 1(5.55) | 0(0) | 0(0) | 1(50) | 1(25) | 1(50) | 24(14.03) |
| Others (10) | 4(5.06) | 1(2.7) | 0(0) | 0(0) | 1(12.5) | 0(0) | 2(25) | 0(0) | 2(50) | 0(0) | 10(5.8) |
| Total | 79(46.19) | 37(21.6) | 6(3.5) | 18(10.5) | 8(4.67) | 7(4.09) | 8(4.67) | 2(1.16) | 4(2.3) | 2(1.16) | 171(100) |

Notes: Others- (ventilators, phototherapy machines, Electro-surgical unit generators, pulse oximeter); Environmental surfaces- (floors, walls, door knobs); n- number of tested strains; %-percentage; CoNS- Coagulase-negative *Staphylococci*.

## Antimicrobial resistance patterns of gram-negative bacteria

The majority of gram-negative bacteria exhibited significantly high resistance to most of the tested antibiotics. For instance, resistance rates were 67.3% for ampicillin, 61.2% for amoxicillin and clavulanic acid, 63.2% for ciprofloxacin, 63.2% for sulfamethoxazole-trimethoprim, 57.1% for cefepime, and 55.1% for piperacillin-tazobactam. Similarly, considerable resistance was observed for amikacin (53.06%), chloramphenicol (53.06%), gentamicin (46.9%), ceftriaxone (34.6%), and ertapenem (32.6%). Lower resistance levels were noted for ceftriaxone (30.6%), cefotaxime (16.3%), and meropenem (16.3%). The highest resistance levels in *Acinetobacter* spp. were seen against penicillin, cephalosporins, carbapenems, and monobactams, with resistance rates of 100% for ampicillin, ceftazidime, and amoxicillin and clavulanic acid, 97.4% for ceftriaxone, and 92.3% for cefotaxime. *Acinetobacter* spp. showed a relatively low resistance to amikacin (35.9%) (Table 3).

The antibiotic resistance rate among gram-positive bacteria was notably high, with 100% resistance to penicillin, azithromycin, clindamycin, and erythromycin. A low resistance level was observed for gentamicin (33.3%) and sulfamethoxazole-trimethoprim (33.3%). Using cefoxitin as a surrogate marker, 50% (n=3/6) of *S. aureus* isolates were identified as MRSA (Table 4).

## MDR level of gram-negative and gram-positive bacteria

Among the 55 bacterial isolates selected for antimicrobial susceptibility testing, multidrug resistance (defined as resistance to at least one antibiotic from three or more different classes) was observed in 61.8% (n=34/55) of the isolates (Table 5).

**Table 3. AMR pattern for gram negative bacteria isolated from environmental surfaces and medical equipment at Zewditu Memorial Hospital.**

| Isolated bacteria | AST pattern | AMI n(%) | AMC n(%) | GEN n(%) | AMP n(%) | CTX n(%) | CTZ n(%) | CTR n(%) | CMP n(%) | CIP n(%) | CPM n(%) | ERT n(%) | MEM n(%) | SXT n(%) | PPT n(%) |
|---|---|---|---|---|---|---|---|---|---|---|---|---|---|---|---|
| P. aeruginosa (18) | R | 17 (94.4) | 11(61.1) | 9(50) | 10(55.5) | 2(11.1) | 5(27.7) | 3(16.6) | 6(33.3) | 6(33.3) | 9(50) | 7(38.8) | 0(0) | 12(66.6) | 9(50) |
|  | S | 1(5.5) | 7(38.88) | 9(50) | 8(44.4) | 16(88.8) | 13(72.2) | 15(83.3) | 12(66.6) | 12(66.6) | 9(50) | 11(61.1) | 18(100) | 6(33.3) | 9(50) |
| E. coli (7) | R | 3(42.8) | 6(85.7) | 3(42.8) | 6(85.7) | 2(28.5) | 3(42.8) | 2(28.5) | 4(57.1) | 6(85.7) | 3(42.8) | 2(28.5) | 0(0) | 4(57.1) | 3(42.8) |
|  | S | 4(57.1) | 1(14.28) | 4(57.1) | 1(14.28) | 5(71.4) | 4(57.1) | 5(71.4) | 3(42.8) | 1(14.28) | 4(57.1) | 5(71.4) | 7(100) | 3(42.8) | 4(57.1) |
| E. cloacae (4) | R | 2(50) | 4(100) | 4(100) | 4(100) | 0(0) | 2(50) | 2(50) | 3(75) | 4(100) | 4(100) | 1(25) | 0(0) | 3(75) | 2(50) |
|  | S | 2(50) | 0(0) | 0(0) | 0(0) | 4(100) | 2(50) | 2(50) | 1(25) | 0(0) | 0(0) | 3(75) | 4(100) | 1(25) | 2(50) |
| Acinetobacter spp. (8) | R | 1(12.5) | 3(37.5) | 2(25) | 6(75) | 3(37.5) | 4(50) | 4(50) | 7(87.5) | 3(37.5) | 4(50) | 2(25) | 2(25) | 4(50) | 5(62.5) |
|  | S | 7(87.5) | 5(62.5) | 6(75) | 2(25) | 5(62.5) | 4(50) | 4(50) | 1(12.5) | 5(62.5) | 4(50) | 6(75) | 6(75) | 4(50) | 3(37.5) |
| Citrobacter spp. (2) | R | 1(50) | 1(50) | 1(50) | 0(0) | 0(0) | 1(50) | 1(50) | 1(50) | 2(100) | 1(50) | 0(0) | 0(0) | 1(50) | 1(50) |
|  | S | 1(50) | 1(50) | 1(50) | 2(100) | 2(100) | 1(50) | 1(50) | 1(50) | 0(0) | 1(50) | 2(100) | 2(100) | 1(50) | 1(50) |
| K. pneumoniae (3) | R | 0(0) | 1(33.33) | 0(0) | 1(33.33) | 0(0) | 0(0) | 0(0) | 1(33.330) | 3(100) | 0(0) | 0(0) | 2(66.66) | 0(0) | 1(33.33) |
|  | S | 3(100) | 2(66.66) | 3(100) | 2(66.66) | 3(100) | 3(100) | 3(100) | 2(66.66) | 0(0) | 3(100) | 3(100) | 1(33.3) | 3(100) | 2(66.66) |
| K. oxytoca (3) | R | 1(33.33) | 1(33.33) | 1(33.33) | 3(100) | 1(33.33) | 1(33.33) | 2(66.66) | 2(66.66) | 3(100) | 3(100) | 1(33.33) | 1(33.33) | 3(100) | 2(66.66) |
|  | S | 2(66.66) | 2(66.66) | 2(66.66) | 0(0) | 2(66.66) | 2(66.66) | 1(33.33) | 1(33.33) | 0(0) | 0(0) | 2(66.66) | 2(66.66) | 0(0) | 1(33.33) |
| K. ozaenae (2) | R | 1(50) | 2(100) | 2(100) | 2(100) | 0(0) | 1(50) | 1(50) | 1(50) | 2(100) | 2(100) | 1(50) | 1(50) | 2(100) | 2(100) |
|  | S | 1(50) | 0(0) | 0(0) | 0(0) | 2(100) | 1(50) | 1(50) | 1(50) | 0(0) | 0(0) | 1(50) | 1(50) | 0(0) | 0(0) |
| Shigella spp (2) | R | 0(0) | 1(50) | 1(50) | 1(50) | 0(0) | 0(0) | 0(0) | 1(50) | 2(100) | 2(100) | 2(100) | 0(0) | 2(100) | 2(100) |
|  | S | 2(100) | 1(50) | 1(50) | 1(50) | 2(100) | 2(100) | 2(100) | 1(50) | 0(0) | 0(0) | 0(0) | 2(100) | 0(0) | 0(0) |

n- number; R-resistant; S-sensitive; %- percentage; AST- antimicrobial susceptibility test; AMI- amikacin; AMC- amoxicillin-clavulanic acid (AMC); GEN- gentamicin; AMP- ampicillin; CTX- cefotaxime; CTZ- ceftazidime; CTR- ceftriaxone; CMP- chloramphenicol; CIP- ciprofloxacin; CPM- cefepime; ERT- ertapenem; MEM- meropenem; SXT- trimethoprim-sulfamethoxazole (SXT); PPT- piperacillin-tazobactam.

**Table 4. AMR pattern for gram-positive bacteria isolated from environmental surfaces and medical equipment at Zewditu Memorial Hospital.**

| Isolated bacteria | AST Pattern | GEN n(%) | CIP n(%) | OXA n(%) | SXT n(%) | PEN n(%) | AZI n(%) | CLI n(%) | ERY n(%) | TET n(%) |
|---|---|---|---|---|---|---|---|---|---|---|
| *S. aureus* | R | 2(33.33) | 3(50) | 3(50) | 2(33.33) | 6(100) | 6(100) | 6(100) | 6(100) | 1(16.66) |
| | S | 4(66.66) | 3(50) | 3(50) | 4(66.66) | 0(0) | 0(0) | 0(0) | 0(0) | 5(83.33) |

n- Number of tested strains; R-resistant; S-sensitive; %- percentage; GEN- gentamicin; CIP-ciprofloxacin; OXA-oxacillin; SXT- sulfamethoxazole+tri-methoprim; PEN- penicillin; AZI- azithromycin; CLI- clindamcin; ERY- erythromycin; TET- tetracycline.

**Table 5. Multi-drug resistance pattern of bacteria isolated from environmental surfaces and medical equipment at Zewditu Memorial Hospital.**

| Gram negative bacteria | Isolated bacteria | $R_0$ n(%) | $R_1$ n(%) | $R_2$ n(%) | $R_3$ n(%) | $R_4$ n(%) | $R_5$ n(%) | | MDR |
|---|---|---|---|---|---|---|---|---|---|
| | *P. aeruginosa* (18) | 5(27.77) | 1(5.55) | 1(5.55) | 4(22.22) | 0(0) | 6(50) | | 10(55.55) |
| | *E. coli* (7) | 1(14.2) | 0(0) | 2(28.5) | 1(14.2) | 0(0) | 3(42.8) | | 4(57.14) |
| | *E. cloacae* (4) | 0(0) | 0(0) | 0(0) | 1(25) | 0(0) | 3(75) | | 4(100) |
| | *Acinetobacter spp.* (8) | 1(12.5) | 1(12.5) | 2(25) | 1(12.5) | 1(12.5) | 2(25) | | 4(50) |
| | *Citrobacter spp.* (2) | 0(0) | 1(33.33) | 0(0) | 0(0) | 0(0) | 1(33.33) | | 1(33.33) |
| | *K. pneumoniae* (3) | 0 (0) | 1(33.33) | 1 (33.33) | 1(33.33) | 0(0) | 0(0) | | 1(33.33) |
| | *K. oxytoca* (3) | 0 (0) | 0 (0) | 0 (0) | 1(33.33) | 1(33.33) | 1(33.33) | | 3(100) |
| | *K. ozanae* (2) | 0(0) | 0(0) | 0(0) | 0(0) | 1(33.33) | 1(33.33) | | 2(66.66) |
| | *Shigella spp.* (2) | 0(0) | 0(0) | 0(0) | 1(33.33) | 0(0) | 1(33.33) | | 2(66.66) |
| Gram positive bacteria | *S. aureus* (6) | 0(0) | 0(0) | 3(50) | 1(16.66) | 0(0) | 1(16.66) | 1(16.66) | 3(50) |

n- Number of tested strains; %.-Percentage; $R_0$- sensitive to all classes of antibiotics; $R_1$-resistant to one class of antibiotics; $R_2$- resistant to two classes of antibiotics; $R_3$- resistant to three classes of antibiotics; $R_4$- resistant to four classes of antibiotics; $R_5$- resistant to five classes of antibiotics; $R_6$- resistant to six classes of antibiotics; MDR- multidrug resistant.

## Extended-spectrum β-lactamase (ESBL) producing bacteria

A total of 18 isolates were suspected to be ESBL producers from 49 Gram-negative bacteria identified. These included *P. aeruginosa* 27.7% (n=5/18), *Acinetobacter spp.* 22.2% (n=4/18), *E. coli* 16.6% (n=3/18), *E. cloacae* 11.1% (n=2/18), *K. oxytoca* 11.1% (n=2/18), *K. ozaenae* 5.5% (n=1/18) and *Citrobacter spp.* 5.5% (n=1/18). Using the combination disc test, the overall prevalence of ESBL-producing bacteria was 26.5% (n=13/49). The breakdown of ESBL positivity by species was as follows: *K. ozaenae* 100% (n=1/1), *E. coli* 66.6% (n=2/3), *K. oxytoca* 50% (n=1/2), *Acinetobacter spp.* 50% (n=2/4), *E. cloacae* 50% (n=1/2), *Citrobacter spp.* 100% (n=1/1), and *P. aeruginosa* 80% (n=4/5) (Fig 2).

The double disk synergy method, another phenotypic confirmatory technique, was used to test all 18 isolates for ESBL production. According to this method, 24.4% (n=12/49) of ESBL cases were confirmed. Among the 13 isolates that tested positive using the combination disc test (CDT), 7.69% (n=1/13) were negative, while 92.3% (n=12/13) were positive. The following bacteria were confirmed as ESBL-positive using the double disk method: *E. coli* 33.33% (n=1/3), *K. ozaenae* 100% (n=1/1), *K. oxytoca* 100% (n=1/1), *Acinetobacter spp.* 50% (n=2/4), *P. aeruginosa* 80% (n=4/5), *E. cloacae* 100% (n=2/2), and *Citrobacter spp.* 100% (n=1/1). Among the bacteria initially suspected of producing ESBL, only 33.3% (n=6/18) were confirmed as ESBL-negative using this approach (Fig 3).

## Carbapenemase-producing bacteria

Out of the 49 bacterial isolates, 18.2% (n=6/49) exhibited intermediate or resistant patterns to imipenem and/or mero-penem, suggesting possible carbapenemase production. These isolates were further tested for carbapenemase production and confirmed phenotypically using the Modified Carbapenemase Inactivation Method (MCIM). The suspected

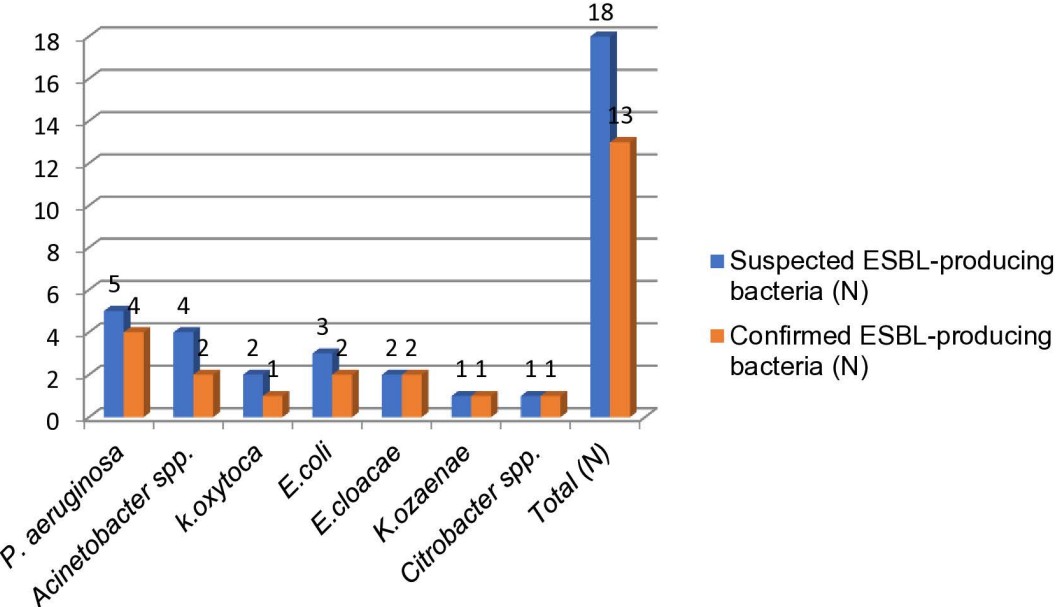

**Fig 2. ESBL-producing bacteria confirmed with combination disk method isolated from environmental surfaces and medical equipment at Zewditu Memorial Hospital.** 7N-number; ESBL-extended-spectrum beta-lactamases.

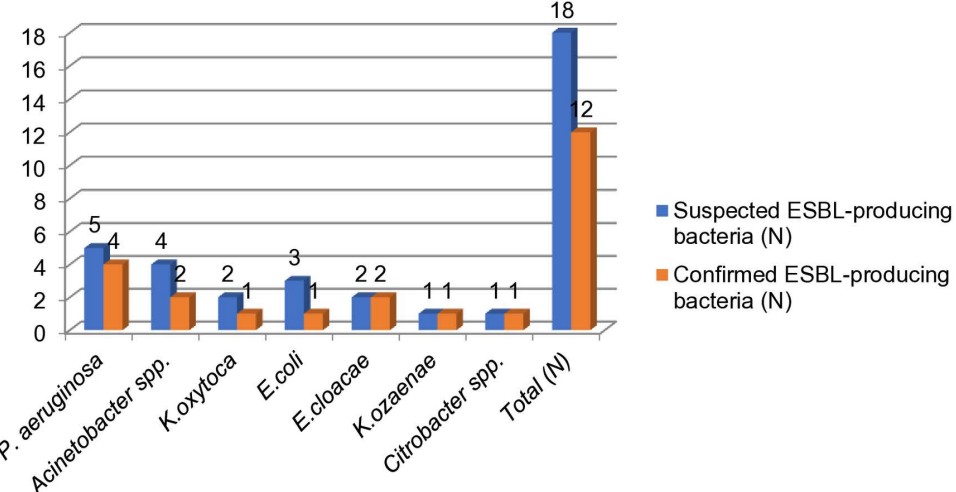

**Fig 3. ESBL-producing bacteria confirmed with Double disk synergy method isolated from environmental surfaces and medical equipment at Zewditu Memorial Hospital.** 8N-number; ESBL-extended-spectrum beta-lactamases.

carbapenem-resistant isolates included *K. pneumoniae* 33.3% (n=2/6), *Acinetobacter* spp. 33.3% (n=2/6), *K. ozaenae* 16.6% (n=1/6) and *K. oxytoca* 16.6% (n=1/6). Of the 6 isolates that showed resistance or intermediate zones for mero-penem, 100% (n=6/6) were positive for carbapenemase production using the MCIM, resulting in an overall prevalence of 12.22% (n=6/49) for carbapenemase-producing bacteria among the total gram-negative isolates. The predominant carbapenemase-producing organisms were *Klebsiella* species, accounting for 8.16% (n=4/49), with *K. pneumoniae* 66.6%

(n=2/3), *K. ozaenae* 50% (n=1/2), and *K. oxytoca* 33.3% (n=1/3) making up the majority. The second most common carbapenemase producer was *Acinetobacter* spp., which accounted for 25% (n=2/8) (Fig 4).

**Methicillin-resistant *Staphylococcus aureus* (MRSA)**

Out of the 6 *S. aureus* isolates, 50% (n=3/6) were identified as MRSA using cefoxitin as a surrogate marker. These confirmed MRSA cases were distributed as follows: 33.33% (n=1/3) in the adult intensive care unit (AICU), 33.33% (n=1/3) in the neonatal intensive care unit (NICU), and 33.33% (n=1/3) in the cesarean section operating room (CS OR) (Fig 5).

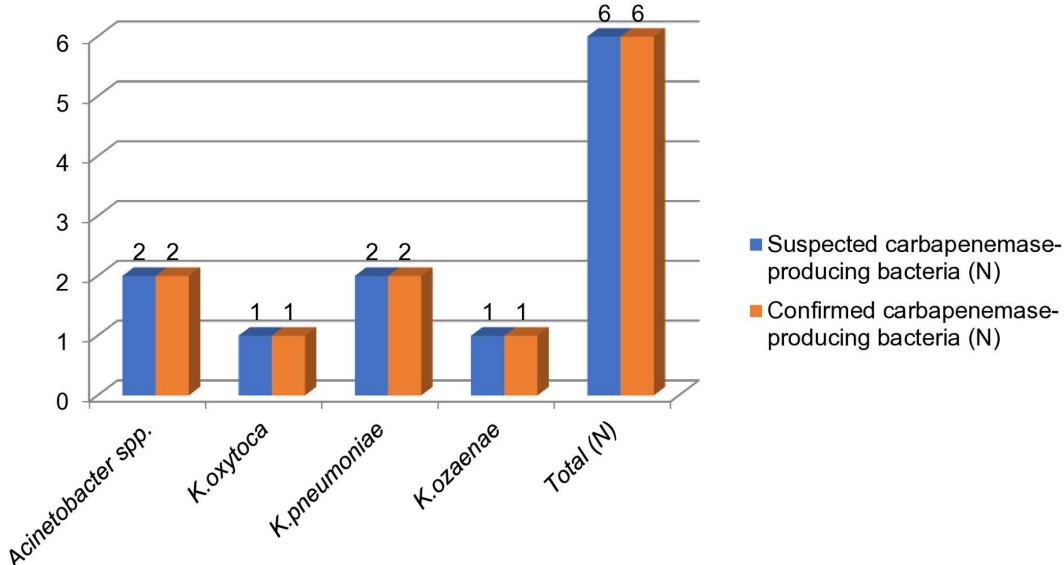

**Fig 4. Carbapenemase-producing bacteria confirmed with MCIM isolated from environmental surfaces and medical equipment at Zewditu Memorial Hospital.** N-number.

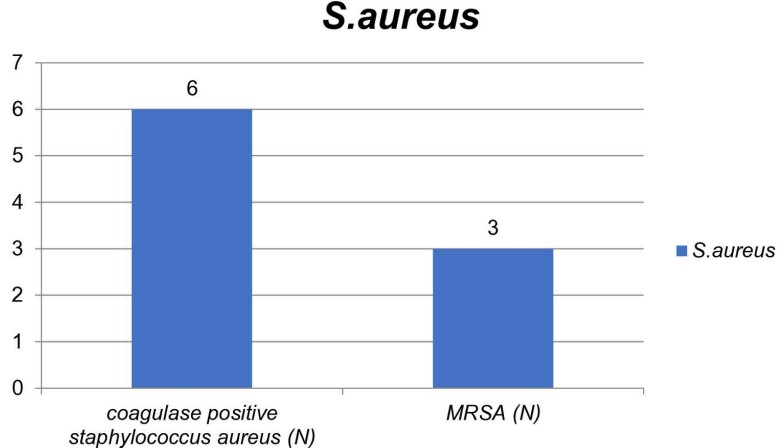

**Fig 5. Methicillin Resistance *S. aureus* (MRSA) isolated from environmental surfaces and medical equipment at Zewditu Memorial Hospital.** N-number.

## Discussion

A previous study has employed molecular typing to examine the clonal relationships between clinical strains and inanimate surfaces, revealing that hospital environments, healthcare workers' (HCWs') hands, and clinical specimens from admitted patients are sources of nosocomial infections [16]. Furthermore, if terminal cleaning is ineffective, patients admitted to rooms previously occupied by individuals infected or colonized with multidrug-resistant (MDR) strains may be up to three times more likely to contract healthcare-associated infections (HCAIs) from contaminated environmental surfaces or equipment [17].

In this study, 158 out of 204 environmental specimens (77.4%) tested positive for bacterial growth and overall, 171 bacterial isolates were found. This finding was consistent with results from similar studies, including a study from Tikur Anbessa Hospital (86%) [18], a nearby hospital to the study site, and another from Mekelle in Northern Ethiopia (88.5%) [19]. A similar result was reported in a study conducted in Nigeria (70.3%) [20]. Other studies from Cameroon (50.4%) [21], Nigeria (46.3.4%) [22], and India (43.3%) [23] reported much lower bacterial contamination rates. These differences could be attributed to variations in ventilation systems, hand hygiene practices, and sterilization and disinfection methods.

The higher bacterial contamination levels observed in this study may be due to ineffective disinfectants, poor adherence to basic precautions like hand washing and contact precautions, and the movement of microorganisms through airflow. Hospitals with inadequate waste management, insufficient awareness of contamination levels, ineffective disinfectants, and a reluctance to invest in contamination control measures (such as proper ventilation systems) are all strongly linked to this situation [24].

These findings indicate substantial contamination of inanimate objects by both Gram-positive (71.3%) and Gram-negative (28.7%) bacteria. Similar results were reported in other studies, such as one conducted at Tikur Anbessa Hospital in Addis Ababa, which found a distribution of 56.3% Gram-positive and 43.7% Gram-negative bacteria [18], as well as studies from Korea (73.2% vs 26.8%) [25], India (56.99% vs 43.01%) [26], and Nigeria (52.2% vs 47.8%) [27]. The prevalence of Gram-positive bacteria could be due to their ability to survive in abiotic hospital environments for several days to months, as they lack the lipid-dominant, desiccation-prone outer membrane found in Gram-negative bacteria [18]. However, the growth and resistance patterns of Gram-positive bacteria are diverse [28]. In contrast, studies conducted in India [23], Indonesia [29], and Morocco [30] found Gram-negative bacteria to be the predominant environmental isolates. These discrepancies may result from differences in sampling times, methods, and culture techniques, as well as variations in the items sampled.

In this study, Coagulase-negative *Staphylococci* (CoNS) were the most frequently isolated bacteria (46.1%), followed by *Bacillus* spp. (21.6%) and *P. aeruginosa* (10.5%) across the wards, which is consistent with findings from various studies worldwide [25, 26]. Although CoNS are part of the normal human skin flora, they can cause clinically significant infections, such as bloodstream and tissue infections [31]. Risk factors for CoNS infections include immune impairment and the use of prosthetic materials, such as intravascular catheters [32].

Among the many surfaces and inanimate objects analyzed in this study, the highest levels of bacterial contamination were found on tables, environmental surfaces, monitors, and bed linens, which aligns with prior research conducted in Ethiopia and other countries [18,33,34,35]. Environmental surfaces and tables were primarily contaminated by CoNS (29.1% and 45.45%), *Bacillus* spp. (37.5% and 18.18%), and *P. aeruginosa* (8.33% and 22.7%), respectively. Similar findings were observed for linens and bed samples in studies from Mizan Tepi, Ethiopia [33]. Cross-contamination from patient flora, healthcare workers' hands, or contaminated patient and healthcare worker footwear may be the source of this contamination.

*Klebsiella spp.* were predominantly found in both adult and neonatal intensive care units (ICUs), where they are associated with ventilator-associated pneumonia and bloodstream infections. A similar result was documented in a study conducted in Iran [36]. The majority of Gram-negative bacteria (GNB) in this study showed high resistance to most tested antibiotics, including ampicillin (97.5%), ceftriaxone (91.3%), ceftazidime (91.3%), amoxicillin-clavulanic acid (85%),

cefotaxime (83.8%), and cefoxitin (76.3%). This high resistance is likely due to the distinctive structure of Gram-negative bacteria. Those results align with similar resistance levels observed in studies from Ethiopia and other African nations, including Tikur Anbessa [18] and Sudan [37]. The high resistance to β-lactam antibiotics is likely caused by selective pressure exerted by these antibiotics [38]. Since these antibiotics are commonly prescribed, their overuse presents significant challenges [18]. On the other hand, resistance to non-β-lactam antibiotics, such as ciprofloxacin (30.6%), meropenem (16.3%), and cefotaxime (16.3%), was lower. This is consistent with findings from Sudan, where ciprofloxacin resistance was 42.7% and amikacin resistance was 23.5% [37].

Gram-positive bacteria in this study exhibited high resistance to azithromycin (100%), penicillin (100%), clindamycin (100%), and erythromycin (100%). Notably, 50% of *Staphylococcus aureus* isolates were methicillin-resistant. This high resistance is consistent with a meta-analysis study in Ethiopia, which reported pooled resistance rates of 97.2% for erythromycin and 99.1% for penicillin [39]. A similar level of penicillin resistance (92.8%) was found at Tikur Anbessa Hospital [18].

Extended-spectrum beta-lactamase (ESBL) production by nosocomial pathogens, such as *E. coli*, *K. pneumoniae*, *A. baumannii*, *P. aeruginosa*, and *Enterobacter* spp., is concerning due to its association with increased hospital costs, prolonged hospital stays, and higher mortality rates [40]. In this study, ESBL-producing bacteria were phenotypically confirmed using the combination disk and double-disc synergy methods. The overall prevalence of ESBL bacteria was 26.5% (n=13/49), which was higher than other studies, such as one by Muzslay et al. (3.1%) [41], Tikur Anbessa Hospital (15.3%), and 9% [42,43], and 20.1% in Tanzania [44]. These differences could be attributed to variations in patient numbers, sample sizes, methodologies, and geographic differences.

This study found that *P. aeruginosa*, *Klebsiella* spp., *E. coli*, and *Acinetobacter* spp. exhibited ESBL production. This was consistent with findings from Tikur Anbessa Hospital, where *Klebsiella* spp. was the most common ESBL producer, followed by *Acinetobacter* spp. and *E. coli* [42]. In Nepal, *Acinetobacter* spp. and *E. coli* were reported as dominant ESBL producers [45], with differences possibly due to variations in the prevalence of Gram-negative bacteria and hygiene practices.

The highest contamination of ESBL-producing bacteria was found on environmental surfaces, followed by oxygen cylinders. In contrast, another study found that the highest contamination occurred on chairs, pooled samples, and sinks [42], with differences likely due to cleaning practices at the hospitals.

Carbapenemase-producing bacterial infections are a significant cause of morbidity and mortality. Carbapenem resistance has been reported in various locations worldwide, including Korea [46]. In this study, carbapenemase-producing bacteria, including *K. ozaenae*, *K. pneumoniae*, *K. oxytoca*, and *Acinetobacter* spp., were analyzed. The current study found that these bacteria were 100% resistant to meropenem, indicating a serious concern regarding carbapenem drugs. Carbapenemase-producing isolates often develop resistance to other antibiotic classes due to gene transfer or mutations in genetic loci [47, 48]. The prevalence of carbapenemase-producing bacteria in this study was 12.2%, which is higher than a study in Korea (0.4%) [46]. The difference may be due to geographical variation, sample sizes, and differences in hygiene practices.

*S. aureus* is a well-known nosocomial pathogen linked to numerous clinical issues in ICUs and operating rooms. In this study, ICUs were most frequently contaminated with *S. aureus*, and 50% of the isolates were MRSA. The overall prevalence of MRSA was 2.45%, which was significantly lower than studies conducted in Ethiopia (85.7%) [18], Nepal (33.3% and 54.4%) [24, 49], and Tanzania (19.5%) [50]. This could be due to differences in sample sizes, hygiene practices, and geographical factors. Compared to other locations, the highest number of *S. aureus* isolates was found on bed linens and bedrails. These surfaces are frequently touched by patients, visitors, and healthcare professionals. Patients in ICUs and operating rooms are particularly vulnerable to nosocomial infections. MRSA and *S. aureus* contamination on these surfaces increases the risk of transmission and can lead to conditions such as pneumonia and septicemia. MRSA-associated infections are difficult to treat due to limited treatment options, highlighting the need for further research in this area.

Hospital design and hygiene practices should primarily focus on managing nosocomial pathogens and resistant strains, which can contaminate surfaces, hands, equipment, and air. A deeper understanding of bacterial contamination mechanisms is essential for developing evidence-based prevention strategies [19]. This study's findings underscore the importance of raising awareness among infection control teams and healthcare workers regarding bacterial contamination in ICUs and operating rooms and its potential link to nosocomial infections. However, the results may not be generalizable due to the study being conducted at a single center, and the relationship between nosocomial infections and contaminated objects or devices was not explored. Therefore, further research in this field is needed.

## Conclusion

*Acinetobacter* spp., *P. aeruginosa* and *S. aureus* were the most commonly identified pathogenic bacteria from inanimate objects at Zewditu Memorial Hospital, all of which are potential causes of healthcare-associated infections. The antimicrobial resistance profile of these isolates was found to increase in clean, inanimate hospital environments. This study highlights a concerning level of multidrug-resistant bacteria, including ESBL and carbapenemase producers, contamination on hospital surfaces. These findings underscore the urgent need for robust infection prevention measures and continuous surveillance of the hospital environment to prevent the spread of resistant bacteria. This is especially important for vulnerable patient groups, such as neonates and those in intensive care units (ICUs) and operating rooms (ORs). Furthermore, advanced research is needed to explore the clonal relationship between clinical strains and inanimate surfaces.

## Supporting information

**S1 File. S1_File.**
(XLSX)

## Acknowledgments

We would like to express our sincere gratitude to Zewditu Memorial Hospital and Tikur Anbessa Specialized Teaching Hospital for allowing us to conduct this study.

## Author contributions

**Data curation:** Hiwot Lidet Yosef.

**Formal analysis:** Hiwot Lidet Yosef.

**Investigation:** Hiwot Lidet Yosef.

**Methodology:** Melese Hailu Legese.

**Supervision:** Melese Hailu Legese.

**Validation:** Melese Hailu Legese.

**Writing – original draft:** Hiwot Lidet Yosef.

**Writing – review & editing:** Melese Hailu Legese, Regasa Diriba, Meron Yohannes.

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
