## [Decision Letter · Decision Letter 0]

5 May 2025

Dear Dr. Yosef,

Thank you for submitting your manuscript to PLOS ONE. After careful consideration, we feel that it has merit but does not fully meet PLOS ONE’s publication criteria as it currently stands. Therefore, we invite you to submit a revised version of the manuscript that addresses the points raised during the review process.

We look forward to receiving your revised manuscript.

Kind regards,

Tsegaye Alemayehu, Msc

Academic Editor

PLOS ONE

2. Please provide additional details regarding participant consent. In the ethics statement in the methods and online submission information, please ensure that you have specified (1) whether consent was informed and (2) what type you obtained (for instance, written or verbal, and if verbal, how it was documented and witnessed). If your study included minors, state whether you obtained consent from parents or guardians. If the need for consent was waived by the ethics committee, please include this information.   

3. In the online submission form, you indicated that [The present study data can be obtained from the author when requested reasonably. Hiwot Lidet: email: halleljezu@gmail.com].

5. Please ensure that you refer to Figure 3, 4, and 5 in your text as, if accepted, production will need this reference to link the reader to the figure.

Additional Editor Comments (if provided):

Reviewers' comments:

Reviewer's Responses to Questions

**Comments to the Author**

1. Is the manuscript technically sound, and do the data support the conclusions?

Reviewer #1: Yes

Reviewer #2: Yes

Reviewer #3: Yes

2. Has the statistical analysis been performed appropriately and rigorously?

Reviewer #1: Yes

Reviewer #2: Yes

Reviewer #3: Yes

3. Have the authors made all data underlying the findings in their manuscript fully available?

Reviewer #1: Yes

Reviewer #2: Yes

Reviewer #3: Yes

4. Is the manuscript presented in an intelligible fashion and written in standard English?

Reviewer #1: Yes

Reviewer #2: Yes

Reviewer #3: Yes

Reviewer #1: Dear author,

1- References 9, 21, 37, and 40 are old, please use new references.

2- Please insert the suitable footnotes for Figures2, 3 and 5.

3- Please insert the suitable footnotes for Tables 3, 4 and 5.

4- Do you have information about false positives or false negatives of samples?

Kind regards

Reviewer #2: Manuscript Number: PONE-D-25-12506

Title: Multidrug-resistant bacterial profiles of inanimate objects at Zewditu

Memorial Hospital, Addis Ababa, Ethiopia.

Overview and general recommendation:

This subject is very important because in developing countries, there is a lack of

information on the extent and types of contamination, as well as the

microbiological profiles of commonly used medical instruments and inanimate

surfaces in hospitals. Therefore, ongoing research in this area is necessary to

address and mitigate the problem.

I found the manuscript well written, well described and the references were new. I

listed minor points below for editing.

Comments to Authors:

Minor points

1- In this manuscript: you wrote (12 times Our)!!! The rule of manuscript

writing is to avoid using (Our). So you should delete (Our), use the formal

scientific words (This review or The current review or The present

review).

2- In 364 line; you wrote (We)!!! The rule of manuscript writing is to avoid

using (We). So you should delete (We), use the formal scientific words

(This review or The current review or The present review).

3- In discussion: 366line; I very recommend you to add new references here

related with [Hospital-acquired infections and carbapenemase-producing

isolates with gene transfer or mutations] to be stronger scientifically. Kindly

add the two mentioned new references

Hasan SA, Raoof WM, Ahmed KK. Antibacterial activity of deer musk and

Ziziphus spina-christi against carbapebem resis-tant gram negative bacteria

isolated from patients with burns and wounds. Regulatory Mechanisms in

Biosystems. 2024 Apr 17;15(2):267-78.

[DOI: https://doi.org/10.15421/022439]

Hasan SA, Raoof WM, Ahmed KK. FIRST REPORT OF CO-

HARBORING BLEOMYCIN RESISTANCE GENE (bleMBL) AND

CARBAPENEMASE RESISTANCE GENE (blaNDM-1) KLEBSIELLA

PNEUMONIAE IN IRAQ WITH COMPARISON STUDY AMONG THE

SENSITIVITY TEST, THE BD PHOENIX CPO DETECT TEST, AND

THE RAPIDEC® CARBA NP TEST. Siberian Journal of Life Sciences and

Agriculture. 2024 Aug 31;16(4):208-37. [https://doi.org/10.12731/2658-6649-2024-16-4-1249]

Kind regards

Reviewer #3: This study presents Multidrug-resistant bacterial profiles of inanimate objects at Zewditu Memorial Hospital, Addis Ababa, Ethiopia. This finding is relevant for clinical and diagnostic applications.

Comments

Line: 72- 73 Common ESBL- and carbapenemase-producing bacteria in nosocomial infections include Escherichia coli, Klebsiella pneumoniae, P. aeruginosa, and Enterobacter cloacae. Reference is required

Line: 83-84 Hospital design and hygiene practices should primarily focus on managing nosocomial pathogens and resistant strains, which can contaminate surfaces, hands, equipment, and air. This sentence is not required here, please remove it. It is better to use it in the discussion part.

Line: 109: ……………ISO/DIS 14698-1 guidelines. Please cite the guideline

Line 120-123. ---------------------CLSI 34th edition (2024). Please cite the CLSI standard

Line: 133----------------------------CLSI guidelines. Please don’t forget to cite while using the guidelines and Literature

Line 170-171: ………. Please remove this sentence and take it with the figure

Table 3: Abbreviations of Antibiotics should be indicated in full form as a footnote in Table 3

Table 3: Are you sure that AMP susceptibility tests are recommended for K. pneumoniae? I don’t think so. Please be sure and correct the typo.

Table 4: What does Ptn stand for in the table? Please write in full form or note in the footnote, please.

Table 4: Please be sure all abbreviations are listed as full forms before being abbreviated, or indicate their full form in the footnote

Lines 24-241: should be removed from here and go with the figure

Line 250-251: Please remember that the topic of figures always goes with the figure. Don’t put it alone in the manuscript. Please take it to the figure.

Line 266-267: Remove the topic and put it in Figure 4

Line 273-274 The topic of the figure is not required here

Line 276: You said that “Several studies ……………..but you only used one reference (14). The term several indicates more than one reference. Could you add more references or modify the sentence, please?

Line 282-283: The rise in antimicrobial resistance (AMR) also contributes to an increased morbidity and mortality rates for HCAIs. This sentence does not make sense. Please rewrite or remove it.

**Do you want your identity to be public for this peer review?** For information about this choice, including consent withdrawal, please see our Privacy Policy

Reviewer #1: No

Reviewer #2: No

Reviewer #3: No

---

## [Author Response · Author response to Decision Letter 1]

21 Jul 2025

Reviewer 1

Reviewer #1: Dear author,

1- References 9, 21, 37, and 40 are old, please use new references.

Author’s response: Thank you very much for your suggestion; we have replaced reference number 9 & 40 and removed reference number 21 and 37.

2- Please insert the suitable footnotes for Figures2, 3 and 5.

Author’s response: Thank you for your comments; we have added the suitable footnotes as you have suggested.

3- Please insert the suitable footnotes for Tables 3, 4 and 5.

Author’s response: Thank you for your comments; we have added the suitable footnotes for our tables as you have suggested.

4- Do you have information about false positives or false negatives of samples?

Author’s response: Thank you for your question; however, there is no information about false positives and false negatives, since the hospital environment samples were processed immediately after collection.

Kind regards

Reviewer #2: Manuscript Number: PONE-D-25-12506

Title: Multidrug-resistant bacterial profiles of inanimate objects at Zewditu Memorial Hospital, Addis Ababa, Ethiopia.

Overview and general recommendation:

This subject is very important because in developing countries, there is a lack of information on the extent and types of contamination, as well as the microbiological profiles of commonly used medical instruments and inanimate surfaces in hospitals. Therefore, ongoing research in this area is necessary to address and mitigate the problem. I found the manuscript well written, well described and the references were new. I listed minor points below for editing.

Comments to Authors:

Minor points

1- In this manuscript: you wrote (12 times Our)!!! The rule of manuscript writing is to avoid using (Our). So you should delete (Our), use the formal scientific words (This review or The current review or The present review).

Author’s response: Thank you very much for your suggestion; we have omitted the word “our’’ and substitute it with relevant scientific words.

2- In 364 line; you wrote (We)!!! The rule of manuscript writing is to avoid using (We). So you should delete (We), use the formal scientific words (This review or The current review or The present review).

Author’s response: Thank you very much for your comment and suggestion; we have omitted the word “We’’ and substitute it with relevant scientific word in line number 366.

2- In discussion: 366line; I very recommend you to add new references here related with [Hospital-acquired infections and carbapenemase-producing isolates with gene transfer or mutations] to be stronger scientifically. Kindly add the two mentioned new references Hasan SA, Raoof WM, Ahmed KK. Antibacterial activity of deer musk and Ziziphus spina-christi against carbapebem resis-tant gram negative bacteria isolated from patients with burns and wounds. Regulatory Mechanisms in Biosystems. 2024 Apr 17;15(2):267-78. [DOI: https://doi.org/10.15421/022439]

Hasan SA, Raoof WM, Ahmed KK. First Report Of Co-Harboring Bleomycin Resistance Gene (Blembl) And Carbapenemase Resistance Gene (Blandm-1) Klebsiella Pneumoniae In Iraq With Comparison Study Among The Sensitivity Test, The Bd Phoenix Cpo Detect Test, And The Rapidec® Carba Np Test. Siberian Journal of Life Sciences and Agriculture. 2024 Aug 31;16(4):208-37. [https://doi.org/10.12731/2658-6649-2024-16-4-1249]

Author’s response: we would like to thank you for your recommendation; we have included those references in line number 388.

Kind regards

Reviewer #3: This study presents Multidrug-resistant bacterial profiles of inanimate objects at Zewditu Memorial Hospital, Addis Ababa, Ethiopia. This finding is relevant for clinical and diagnostic applications.

Comments

Line: 72- 73 Common ESBL- and carbapenemase-producing bacteria in nosocomial infections include Escherichia coli, Klebsiella pneumoniae, P. aeruginosa, and Enterobacter cloacae. Reference is required

Author’s response: Thank you very much for your suggestion; we have added a reference to the line number 72-73.

Line: 83-84 Hospital design and hygiene practices should primarily focus on managing nosocomial pathogens and resistant strains, which can contaminate surfaces, hands, equipment, and air. This sentence is not required here, please remove it. It is better to use it in the discussion part.

Author’s response: Thank you very much for your comment; we have moved the sentence into the discussion part, line number 385-388.

Line: 109: ……………ISO/DIS 14698-1 guidelines. Please cite the guideline

Author’s response: Thank you very much for your comment; we have cited the sentence with reference.

Line 120-123. ---------------------CLSI 34th edition (2024). Please cite the CLSI standard

Author’s response: Thank you very much for your suggestion; we have cited the sentence with reference.

Line: 133----------------------------CLSI guidelines. Please don’t forget to cite while using the guidelines and Literature

Author’s response: Thank you very much for your suggestion; we have cited the sentence with reference.

Line 170-171: ………. Please remove this sentence and take it with the figure

Author’s response: Thank you very much for your suggestion; we have moved the sentence into the figure.

Table 3: Abbreviations of Antibiotics should be indicated in full form as a footnote in Table 3

Author’s response: Thank you very much for your suggestion; we have added the full forms of antibiotics as a footnote in Table 3.

Table 3: Are you sure that AMP susceptibility tests are recommended for K. pneumoniae? I don’t think so. Please be sure and correct the typo.

Author’s response: Thank you very much for your question, but yes CLSI guideline recommends testing AMP for Enterobacteriaceae that includes K. pneumoniae.

Table 4: What does Ptn stand for in the table? Please write in full form or note in the footnote, please.

Author’s response: Thank you very much for your question and suggestion. We have written it in full form in the table.

Table 4: Please be sure all abbreviations are listed as full forms before being abbreviated, or indicate their full form in the footnote

Author’s response: Thank you very much for your suggestion. We have written all abbreviation in full form.

Lines 24-241: should be removed from here and go with the figure

Author’s response: Thank you very much for your suggestion; we have moved the sentence into the figure.

Line 250-251: Please remember that the topic of figures always goes with the figure. Don’t put it alone in the manuscript. Please take it to the figure.

Author’s response: Thank you very much for your suggestion; we have moved the sentence into the figure.

Line 266-267: Remove the topic and put it in Figure 4

Author’s response: Thank you very much for your suggestion; we have moved the sentence into the figure.

Line 273-274 The topic of the figure is not required here

Author’s response: Thank you very much for your suggestion; we have moved the sentence into the figure.

Line 276: You said that “Several studies ……………..but you only used one reference (14). The term several indicates more than one reference. Could you add more references or modify the sentence, please?

Author’s response: Thank you very much for your comment; we have modified the word in the sentence.

Line 282-283: The rise in antimicrobial resistance (AMR) also contributes to an increased morbidity and mortality rates for HCAIs. This sentence does not make sense. Please rewrite or remove it.

Author’s response: Thank you very much for your suggestion; we have removed the sentence.

---

## [Decision Letter · Decision Letter 1]

18 Aug 2025

Dear Dr. Hiwot Lidet Yosef,

We look forward to receiving your revised manuscript.

Kind regards,

Tsegaye Alemayehu, Msc

Academic Editor

PLOS ONE

Journal Requirements:

Reviewers' comments:

Reviewer's Responses to Questions

**Comments to the Author**

Reviewer #1: (No Response)

Reviewer #2: All comments have been addressed

2. Is the manuscript technically sound, and do the data support the conclusions?

Reviewer #1: Yes

Reviewer #2: Yes

3. Has the statistical analysis been performed appropriately and rigorously?

Reviewer #1: Yes

Reviewer #2: Yes

4. Have the authors made all data underlying the findings in their manuscript fully available?

Reviewer #1: Yes

Reviewer #2: Yes

5. Is the manuscript presented in an intelligible fashion and written in standard English?

Reviewer #1: Yes

Reviewer #2: Yes

Reviewer #1: Dear author,

This is a good article and could provide useful information for readers of this article. Please make the following corrections. References 24 and 44 are old, please use new references.

Kind regards

Reviewer #2: Greetings

Very good work.

You did all the scientific requirements by the correct way.

Now, your manuscript is more accurate.

Kind regards.

**Do you want your identity to be public for this peer review?** For information about this choice, including consent withdrawal, please see our Privacy Policy

Reviewer #1: No

Reviewer #2: No

---

## [Author Response · Author response to Decision Letter 2]

25 Aug 2025

Dear Editor; PLOS ONE

Thank you very much for allowing us to revise our manuscript entitled “Multidrug-resistant bacterial profiles of inanimate objects at Zewditu Memorial Hospital, Addis Ababa, Ethiopia.”. We value the reviewers' insightful comments and have thoughtfully considered and responded. Our thorough answer is included in the attachment below. The reviewers' comments have been copied and pasted, and the author's responses are italicized. We have made changes to the references of the article in this revision, which are indicated in the updated manuscript through track changes. We sincerely thank the reviewers for their insightful comments of our article.

With best regards;

Hiwot Lidet;

Corresponding Author

Responses to the reviewers’ comments

Reviewer 1

Reviewer #1: Dear author,

This is a good article and could provide useful information for readers of this article. Please make the following corrections. References 24 and 44 are old, please use new references.

Author’s response: Thank you very much for your comment; we have removed those references since they are old.

Kind regards.

Reviewer 2

Reviewer #2: Greetings

Very good work.

You did all the scientific requirements by the correct way.

Now, your manuscript is more accurate.

Kind regards.

Author’s response: Thank you very much.

---

## [Decision Letter · Decision Letter 2]

14 Sep 2025

Multidrug-resistant bacterial profiles of inanimate objects at Zewditu Memorial Hospital, Addis Ababa, Ethiopia.

PONE-D-25-12506R2

Dear Dr. Yosef,

We’re pleased to inform you that your manuscript has been judged scientifically suitable for publication and will be formally accepted for publication once it meets all outstanding technical requirements.

Kind regards,

Tsegaye Alemayehu, Msc

Academic Editor

PLOS ONE

Additional Editor Comments (optional):

Reviewer #1:

Reviewers' comments:

Reviewer's Responses to Questions

**Comments to the Author**

Reviewer #1: (No Response)

2. Is the manuscript technically sound, and do the data support the conclusions?

Reviewer #1: Yes

3. Has the statistical analysis been performed appropriately and rigorously?

Reviewer #1: Yes

4. Have the authors made all data underlying the findings in their manuscript fully available?

Reviewer #1: Yes

5. Is the manuscript presented in an intelligible fashion and written in standard English?

Reviewer #1: Yes

Reviewer #1: Dear Author,

This article titled "Multidrug-resistant bacterial profiles of inanimate objects at Zewditu Memorial Hospital,

Addis Ababa, Ethiopia." is a good article and provides good information to the readers of this article.

Kind regards

**Do you want your identity to be public for this peer review?** For information about this choice, including consent withdrawal, please see our Privacy Policy

Reviewer #1: No

---

## [Editor Report · Acceptance letter]

PONE-D-25-12506R2

PLOS ONE

Dear Dr. Yosef,

I'm pleased to inform you that your manuscript has been deemed suitable for publication in PLOS ONE. Congratulations! Your manuscript is now being handed over to our production team.

Kind regards,

on behalf of

Dr. Tsegaye Alemayehu

Academic Editor

PLOS ONE